# Evolution of heterogeneity under constant and variable environments

**Ryo Oizumi**[1]*, **Hisashi Inaba**[2]

**1** National Institute of Population and Social Security Research, Tokyo, Japan, **2** Graduate School of Mathematical Science, The University of Tokyo, Tokyo, Japan

* ooizumi-ryou@ipss.go.jp

**Data Availability Statement:** All relevant data are within the manuscript and its Supporting information files.

**Funding:** This research was supported by the Japan Society for the Promotion of Science (JSPS) KAKENHI (Grant number 20K14368) and the

## Abstract

Various definitions of fitness are essentially based on the number of descendants of an allele or a phenotype after a sufficiently long time. However, these different definitions do not explicate the continuous evolution of life histories. Herein, we focus on the eigenfunction of an age-structured population model as fitness. The function generates an equation, called the Hamilton–Jacobi–Bellman equation, that achieves adaptive control of life history in terms of both the presence and absence of the density effect. Further, we introduce a perturbation method that applies the solution of this equation to the long-term logarithmic growth rate of a stochastic structured population model. We adopt this method to realize the adaptive control of heterogeneity for an optimal foraging problem in a variable environment as the analyzable example. The result indicates that the eigenfunction is involved in adaptive strategies under all the environments listed herein. Thus, we aim to systematize adaptive life histories in the presence of density effects and variable environments using the proposed objective function as a universal fitness candidate.

## Introduction

Since the publication of The Origin of Species by Charles Darwin, many biologists have believed that evolution is promoted by mutation and adaptation. Mutation is a well-known phenomenon that has been extensively studied at the molecular level. Similarly, adaptation is a widely accepted idea, and its degree is estimated by an index called "fitness," which has been defined in several ways essentially based on the number of descendants of an allele or a phenotype after a sufficiently long time. If an allele or an individual with a mutation has greater fitness than other alleles/individuals without a mutation, the mutation will eventually dominate the population. However, fitness is not observed easily in nature; therefore, we must rely on indirect indices to analyze evolution.

Because it cannot be easily observed in nature, fitness does not have a unique and quantitative definition. An adaptive gene must meet several requirements to thrive in a population. The indicator must be a measure by which adaptive genes dominate the population, regardless of population dynamics, including saturated growth, exponential growth, or stochastic growth.

Ministry of Health, Labour and Welfare (Grant number 20AA2007).

**Competing interests:** The authors have declared that no competing interests exist.

Biologists use population growth rates, basic reproductive numbers, and abstract payoff functions instead of substantial fitness and often call them "fitness." These indices can represent fitness in restricted environments; e.g., (1) absence of intraspecific and interspecific competition, (2) population dynamics limited to one generation, and (3) negligible population dynamics. However, these conditions are unusual in the natural world. Therefore, the environments surrounding organisms are believed to comprise combinations of these conditions.

For example, for the combination of (1) and (2), we can determine the fitness associated with the life schedules of individuals and the population dynamics. A study addressing this problem linked age-structured models to control theory [1]. The researchers used the characteristic function of the Euler–Lotka equation as the fitness metric. Although this model was constructed to maximize the characteristic function with the adaptive life schedule, it maximized the population growth rate. The maximization of the characteristic function is equivalent to the maximization of population growth. Further, the model provided a framework for the analysis of the adaptive control of life history to natural selection.

The systematization of mathematical models related to the evolution of life histories has been promoted by linking the behavior of individuals to their population dynamics. One of the most challenging aspects of finding a general definition of fitness is that general population dynamics contain intra- and inter-specific competition, which complicates the dynamics and makes it challenging to identify what the species optimizes. The $r/K$ selection theory argues that the adaptive life history in a stationary population maximizes the carrying capacity [2]. Although this argument has long been controversial, it has not revealed a satisfactory strategy through which the life schedule maximizes the carrying capacity.

A recent report proposed that species maximize the common objective function in both $r$-selection and $K$-selection [3]. This function provides the characteristic function of the Euler–Lotka equation—the same as in general studies [1, 4]—but it does not incorporate a parameter such as the carrying capacity. Instead, the function contains a density effect that represents the intra-specific competition from each age and state. If the density effect generates a stationary population, it indicates the carrying capacity and provides an optimal life history in $K$-selection. According to this model, density effects evolve various life histories not only with precocity and prolificacy but also by maximizing the population growth. This phenomenon has been observed in another study [5].

An adaptive condition for species requires not only the maximization of the population growth rate but also an evolutionarily stable strategy (ESS): no mutants can invade the population or the genetic pool. In previous research [4], the carrying capacity was considered a constraint because the objective function was assumed to monotonically decrease in terms of the density effect.

Those studies unified the population growth rate and the basic reproduction number via a characteristic function. The former is not thought to be appropriate for fitness under a saturated population. Conversely, the latter does not always become a larger population than the species maximizing the former because it does not consider the generation time. Maximizing the characteristic function maximizes the population growth rate in $r$–selection and maximizes the basic reproduction number in $K$–selection. In other words, these parameters as fitness are a result of maximizing the characteristic function and not a direct indicator of evolution. The applicability of this framework in a variable environment remains to be determined.

Evolution in a variable environment was established via sensitivity analysis [6] and Tuljapurkar's approximation [7]. These methods have been systematized for general transition-matrix models. Recent studies focused on the effects of these structures on population dynamics in variable environments [8–11]. Each life history changes with age and has individual differences; however, it is not always reasonable to observe the growth of the physiological state

with age in the field research because it is difficult to divide heterogeneity into each age structure in many cases. Therefore, biologists often apply state-structure models without age to their analysis. Essentially, researchers of empirical studies now need to consider each age structure. Evolution cannot ignore age because natural selection is believed to act with individual life histories. Multistate structured models involving age are an increasingly important area of demography and ecology [12–14].

In this study, we construct a method that addresses the adaptive life schedule in the absence and presence of a variable environment based on a continuous multi-state age-structured population model. Our method follows the general theorem for *r/K*-selection established by Oizumi et al. [3] and derives a more generalized control equation for the adaptive life strategy from it in a constant environment. Further, we construct a perturbation method that corresponds to Tuljapurler's approximation in continuous models. We adopt this method for the adaptive control of heterogeneity for an optimal foraging problem in a variable environment as an analyzable example. Next, by comparing adaptive strategies in the presence and absence of a variable environment, we suggest that there exists an adaptive threshold for the variance of heterogeneity under environmental stochasticity. This study systematizes adaptive life histories in the presence of individual heterogeneity, density effects, and environmental stochasticity using the aforementioned objective function.

Our results reveal that fitness is closely related to the reproductive value. We show that characteristic functions play an important role in population dynamics even in constant and variable environments. Our model shows that heterogeneity is more likely to evolve in a variable environment than in a constant environment. Our framework will help us find a universal definition of fitness.

## Theory and mathematical methods

### Multi-state age-structured population model

We developed a general model theory for biomathematics. We define the state-growth model for each trait. Suppose that $y \in A \subseteq \mathbb{R}^d$ are $d$-dimensional trait features characterizing each individual where $A$ is the domain of $y$. The growth of each trait from age $a_0$ to $a$ is assumed to be described by a $d$-dimensional Ito-type diffusion process

$$X_a^j = y^j + \int_{a_0}^{a} g_j(s, X_s)ds + \sum_{\ell=1}^{N} \int_{a_0}^{a} \sigma_{j\ell}(s, X_s)dB_s^\ell \quad j = 1, 2, \cdots, d. \tag{1}$$

$B_\tau^\ell$ represents the $\ell$-th element of the $N$-dimensional Brownian motion and $\sigma_{j\ell}(\cdot)$ comprises

$$S_{j,j'}(a, y) := \sum_{l=1}^{N} \sigma_{\ell j}(a, y)\sigma_{\ell j'}(a, y).$$

Further, $g_j(\cdot)$ and $S_{jj'}(\cdot)$ represent the mean and covariance of $j$-th state growth rates, respectively.

This SDE can be interpreted as a rule for each state growth of individuals. The heterogeneity of individuals generated by the SDE is referred to as internal stochasticity to distinguish it from environmental stochasticity, which is external stochasticity [15].

For the boundary value $x \in \partial A$, each state transition rate and fluctuation term are assumed to be zero (Dirichlet condition). The age-specific fertility rate in state $y$ is given by $F(a, y) \geq 0$,

and the force of mortality is assumed to satisfy

$$\mu \in L^1_{\text{loc},+}([0,\alpha) \times A), \quad \int_0^\alpha da\ \mu(a,y) = \infty \ \forall y \in A, \tag{2}$$

in each state because $\alpha$ denotes the maximum attainable age.

Let the population vector $P_t(a,y)$, in which each individual follows the ingredients Eq (1), $F(a,y)$, and Eq (2), be a cohort density at age $a$ at a state $y$ in time $t$. Then, we obtain the basic partial differential equation as

$$\left(\frac{\partial}{\partial t} + \frac{\partial}{\partial a}\right) P_t(a,y) = -H(a,y)P_t(a,y), \tag{3}$$

where the linear operator $H(a,y)$ is given by [16]

$$H(a,y)\phi(y) = \mu(a,y) + \sum_{j=1}^d \frac{\partial}{\partial y^j}\left(g_j(a,y)\phi(y)\right) - \frac{1}{2}\sum_{\substack{j=1 \\ j'=1}}^d \frac{\partial^2}{\partial y^j \partial y^{j'}}\left(S_{jj'}(a,y)\phi(y)\right).$$

Eq (3) implies that the cohort transitions dynamically for age $a$ and state $y$ at time $t$.

In addition, we assume that the boundary condition representing the birth law is given by

$$P_t(0,y) = v(y) \int_0^\alpha \int_A dadx\ F(a,x)P_t(a,x), \tag{4}$$

where $v(\cdot) \in L^1_+(A)$ represents the state distribution of the neonatal population satisfying

$$\int_A dy\ v(y) = 1.$$

**Basic renewal process.** Let $p_t(a) := P_t(a,\cdot)$ represent the age-density function at time $t$ considering a value in the trait space $E = L^1(A)$; further, let $X := L^1(0,\alpha;E)$ be the state space of the age-density functions. Then, the basic system (Eqs (3) and (4)) is formulated as an abstract McKendrick equation given as

$$\left(\frac{\partial}{\partial t} + \frac{\partial}{\partial a}\right)p_t(a) = -\mathcal{H}(a)p_t(a),$$
$$p(t,0) = \int_0^\alpha da\ \mathcal{F}(a)p_t(a), \tag{5}$$

where $\mathcal{H}(a)$ is a linear operator acting on $E$ given by $(\mathcal{H}(a)f)(y) = H(a,y)f(y)$ for $f \in E$, and $\mathcal{F}(a)$ is a one-dimensional linear operator from $E$, given by

$$(\mathcal{F}(a)f)(y) = v(y)\int_A d\zeta\ F(a,\zeta)f(\zeta), \ f \in E. \tag{6}$$

Suppose that the operator $-\mathcal{H}(a)$ generates an evolutionary system $U(a,s)$, $a \geq s \geq 0$, on $E$. Then, for $\varphi \in D(\mathcal{H}(a))$, it holds that

$$\frac{\partial^+}{\partial a} U(a,s)\varphi|_{a=s} = -\mathcal{H}(s)\varphi,$$
$$\frac{\partial}{\partial s} U(a,s)\varphi = U(a,s)\mathcal{H}(s)\varphi, \tag{7}$$

and the solution $p$ is expressed as

$$p_t(a) = \begin{cases} U(a, a-t)p_0(a-t), & t \le a, \\ U(a, 0)p_{t-a}(0), & t > a. \end{cases} \tag{8}$$

Let $\beta(t) := p_t(0) \in E$ be the density of the newborns at time $t$.

$$\beta(t) = \int_0^\alpha da \ \mathcal{F}(a)p_t(a).$$

Substituting Eq (8) into the boundary condition of Eq (5), we have

$$\beta(t) = G(t) + \int_0^t da \ \Psi(a)\beta(t-a), \tag{9}$$

where

$$\Psi(a) := \mathcal{F}(a)U(a, 0),$$

$$G(t) := \int_t^{\max(t,\alpha)} da \ \mathcal{F}(a)U(a, a-t)v.$$

Then, $\Psi(a)$ is a one-dimensional positive operator on $E$, whose range is spanned by $v$; the next generation operator is $K = \int_0^\alpha da \ \Psi(a)$. Thus, the spectral radius is given by

$$r(K) = \int_0^\alpha da \int_A d\zeta \ F(a, \zeta)(U(a, 0)v)(\zeta), \tag{10}$$

which is the reproduction number $R_0$ of our system.

Let $\hat{\Psi}(\lambda) := \int_0^\alpha da \ \exp\{-\lambda a\}\Psi(a)$ and $r \in \mathbb{C}$. Then, there exists a unique real root $r_0$ satisfying the characteristic equation $\Lambda(\hat{\Psi}(\lambda)) = 1$, i.e.,

$$\int_0^\alpha da \int_A d\zeta \ \exp\{-\lambda a\}F(a, \zeta)(U(a, 0)v)(\zeta) = 1. \tag{11}$$

It follows from the well-known renewal theorem [17, 18] that there exist numbers $C_0 > 0$ and $\eta > 0$ such that

$$\beta(t) = C_0 \exp\{r_0 t\}[1 + O(\exp\{-\eta t\})]. \tag{12}$$

where $r_0$ is known as the dominant characteristic root:

$$r_0 > \Re r_k \ge \Re r_{k+1}, \ \ k = 1, 2 \cdots,$$

and $r_k$ ($k = 0, 1, 2, \cdots$) are the characteristic roots of (11)

$$r_k \in \Lambda := \left\{ \lambda \in \mathbf{C} : \int_0^\alpha da \int_A d\zeta \exp\{-\lambda a\}F(a, \zeta)(U(a, 0)v)(\zeta) = 1 \right\}.$$

The long-term logarithmic growth rate (LLGR) of the population denoted by $\bar{r}$ is defined as

$$\bar{r} := \lim_{t \to \infty} \frac{1}{t} \ln\{\|p_t(\cdot)\|_X\}, \tag{13}$$

where $L^1$-norm $\|\cdot\|_X$ is defined as

$$\|\phi\|_X := \int_0^\alpha da \ |\phi(a)|_E,$$

where $|\cdot|_E$ denotes the $L^1$-norm of the trait space $E$. From the renewal theorem (9), we have $\bar{r} = r_0$ in a constant environment.

## Eigenvalue problem

Let

$$\mathscr{H} := -\frac{d}{da} - \mathcal{H}(a),$$

be a linear operator on $X$ with domain

$$D(\mathscr{H}) = \left\{ \varphi \in X : \mathscr{H}\varphi \in X, \varphi(0) = \int_0^\alpha \mathcal{F}(a)\varphi(a)da \right\}.$$

Then, (5) can be viewed as an ordinary differential equation on the Banach space $X$.

$$\frac{dp_t}{dt} = \mathscr{H}p_t, \tag{14}$$

where $p_t = p_t(\cdot)$ is a population vector taking a value in $X$.

Then, $\mathscr{H}$ becomes an infinitesimal generator of the $C_0$-semigroup $T(t)$, $t \geq 0$, on $X$, and $\mathscr{H}$ has eigenfunctions $w_k$ as

$$w_k(a) = \exp\{-r_k a\}U(a,0)v, \ \ k = 0, 1, 2, \cdots. \tag{15}$$

Consider an adjoint operator $\mathscr{H}^*$ and its eigenfunction of $w_k^*$. Let us introduce the duality pairing $\langle v, w \rangle_X$ between $v \in X^*$ and $w \in X$ as

$$\langle v, w \rangle_X := \int_0^\alpha \int_A dady \ v(a,y)w(a,y).$$

From the relationship $\langle \mathscr{H}^* v, w \rangle = \langle v, \mathscr{H}w \rangle$, we have

$$(\mathscr{H}^* v)(a) := \frac{dv(a)}{da} - \mathcal{H}^*(a)v(a) + v(0)vF(a,\cdot), \tag{16}$$

where the domain is given by

$$D(\mathscr{H}^*) = \{v \in X^* : \mathscr{H}^* v \in X^*, v(\alpha) = 0\}$$

and $\mathcal{H}^*(a)$ is a linear operator on $E^*$ given by

$$\mathcal{H}^*(a) := -\sum_{j=1}^d g_j(a,y)\frac{\partial}{\partial y^j} - \frac{1}{2}\sum_{\substack{j=1 \\ j'=1}}^d S_{jj'}(a,y)\frac{\partial^2}{\partial y^j \partial y^{j'}} + \mu(a,y).$$

The adjoint operator $-\mathcal{H}^*(a)$ is the generator for the adjoint evolutionary system $U^*(a,s) = U(s,a)^*$, $s \geq a$. It follows from (7) that

$$\frac{\partial}{\partial s}U^*(a,s)\phi^* = \mathcal{H}^*(a)U^*(a,s)v, \ \ v \in E^*. \tag{17}$$

It is reasonable to define the adjoint eigenfunction corresponding to the dominant eigenvalue $r_0$ as the *reproductive value*. From the adjoint eigenvalue problem $\mathcal{H}^* v_k = r_k v_k$, we have the adjoint eigenvector associated with the eigenvalue $r_k$ as

$$v_k(a) = \int_a^\alpha ds \ \exp\{-r_k(s-a)\} U^*(a,s) v_k(0) v F(s, \cdot), \tag{18}$$

where $v_k(0)$ is an arbitrary value in $E$.

From a stochastic perspective, transition operators $U$ and $U^*$ are represented by a fundamental solution $K(s, x \to a, y)$ satisfying

$$\begin{aligned} \frac{\partial}{\partial a} K(s, x \to a, y) &= -\mathcal{H}(a) K(s, x \to a, y) \\ K(s, x \to s, y) &= \delta^d(x-y) \end{aligned} \quad (s \le a \le \alpha).$$

or

$$\begin{aligned} \frac{\partial}{\partial s} K(s, x \to a, y) &= -\mathcal{H}^*(s) K(s, x \to a, y) \\ K(a, x \to a, y) &= \delta^d(x-y) \end{aligned} \quad (s \le a \le \alpha)$$

(cf. [19]). Therefore, Eqs (15) and (18) can be rewritten as

$$w_k(a, y) = \exp\{-r_k a\} \int_A dy \ v(x) K(0, x \to a, y) \tag{19}$$

$$v_k(a, y) = \int_A d\xi \ v_k(0, \xi) v(\xi) \int_a^\alpha ds \ \exp\{-r_k(s-a)\} \int_A d\zeta \ K(a, y \to s, \zeta) F(s, \zeta). \tag{20}$$

Accordingly, characteristic Eq (11) becomes

$$\int_0^\alpha da \int_A d\zeta \ \exp\{-\lambda a\} F(a, \zeta) \int_A dy \ v(x) K(0, x \to a, \zeta) = 1.$$

This fundamental solution $K(s, x \to a, y)$ implies the transition probability of the state growth from an initial state $x$ at age $s$ to a final state $y$ at age $a$; this is generated by Eq (1).

Using eigenfunctions, we can obtain an asymptotic expansion of the population semigroup.

$$T(t)\varphi := \sum_{k=0}^n \frac{\langle v_k, \varphi \rangle}{\langle v_k, w_k \rangle} \exp\{r_k t\} w_k + O(\exp\{(\Re r_k - \epsilon)t\}), \tag{21}$$

where $\epsilon$ is a small positive number [20].

Further, it is easy to see that the total reproductive value $V(t) := \langle v_0, T(t)\varphi \rangle$ satisfies

$$V(t) = V(0) \exp\{r_0 t\}, \tag{22}$$

from which we have

$$\bar{r} = r_0 = \lim_{t \to \infty} \frac{\ln\{V(t)\}}{t}. \tag{23}$$

This derivation via functional analysis is technically convenient for defining the semigroup operator using eigenfunctions; further, a stochastic interpretation of those eigenfunctions is reasonable to connect the population dynamics with the life histories of individuals. The latter

interpretation is required to derive the Hamilton–Jacobi–Bellman equation involved in the adaptive control of life history, and we address this later.

## General adaptive life history in a constant environment

To the best of our knowledge, the study of adaptive life histories using structured population models began with [1, 4]. These studies verified that maximizing the characteristic function (Eq (11)) is equivalent to maximizing the dominant characteristic root $r_0$. Further, recent studies have extended this theorem to address internal stochasticity and density effects by adopting the stochastic control theory [3, 16].

Let us consider the general population dynamics containing the control parameter $u \in \mathbb{U} \subset \mathbb{R}^{d'}$, where $u$ represents a value in the given Borel set $\mathbb{U}$ to control each state $X_a$ [21]. If the $d'$-dimensional density effect is given by $\Gamma_t \in \mathbb{R}_+^{d''}$, the general population dynamics are

$$\left(\frac{\partial}{\partial t} + \frac{\partial}{\partial a}\right) P_t(a, y) = -H(a, y, u, \Gamma_t) P_t(a, y). \tag{24}$$

Moreover, the renewal process of this system is given by

$$P_t(0, y) = v(y) \int_0^\alpha da \int_A dadx \ F(a, x, u_a, \Gamma_t) P_t(a, x).$$

Then, if $\gamma_{\ell'} = \gamma_{\ell'}(a, y)$ is a weight function for each age and state, the vector of $d'$-dimensional density effect $\Gamma_t$ is given by

$$\Gamma_t := (\Gamma_t^{\ell'})_{0 \leq \ell' \leq d''}, \ \ \Gamma_t^{\ell'} := \langle \gamma_{\ell'}, P_t \rangle.$$

For simplicity, $H(a, y, u, \Gamma)$ is assumed to be an adjoint Fokker–Planck Hamiltonian parameterized by constant vectors $u$ and $\Gamma$

$$H(a, y, u, \Gamma)\phi(y) :=$$

$$\sum_{j=1}^d \left(\frac{\partial}{\partial y^j} g_j(a, y, u, \Gamma)\phi(y \ right))\right) - \frac{1}{2}\sum_{j'=1}^d \left(\frac{\partial^2}{\partial y^j \partial y^{j'}} S_{j,j'}(a, y, u, \Gamma)\phi(y)\right) \tag{25}$$

$$+\mu(a, y, u, \Gamma)\phi(y)$$

Suppose that fertility depends on states $y$, $u$, and $\Gamma$ such that

$$F(a, y, u, \Gamma_t) = F(a, y, u, \Gamma). \tag{26}$$

These assumptions assume that the density effects are approximated to zero or are constant in sufficiently small or stationary populations.

Here, $\phi[u]$ indicates that $\phi$ is a functional with respect to $u$. If $\tilde{u}(a, X_a) \in \mathbb{U}$ is the adaptive control of the life schedules, it should satisfy the following theorem.

**Theorem 0.1** *Let* $\tilde{u} = \tilde{u}(a, X_a)$ *be*

$$\tilde{u} \in \mathbb{U}, \ s.t. \ r_0[\tilde{u}](\Gamma) = \sup_{u \in \mathbb{U}} r_0[u](\Gamma),$$

*Let* $\psi_r[u](\Gamma)$ *be given by*

$$\psi_r[u](\Gamma) := \int_0^\alpha da \ \exp\{-ra\} F(a, y, u, \Gamma)(U(a, 0, v, \Gamma)v)(y)$$

*and $\psi_{r_0[u]}[u](\Gamma) = 1$. We define $\tilde{r}_0(\Gamma)$ as $\tilde{r}_0 := r_0[\tilde{u}](\Gamma)$ as follows: Then, we have*

$$\psi_{\tilde{r}_0}[u](\Gamma) \leq \psi_{\tilde{r}_0}[\tilde{u}](\Gamma) = 1.$$

This theorem is easily verified because of the monotonicity of $\psi_r[u](\Gamma)$ with respect to $r$. The theorem implies that a control that maximizes $\psi_r[u](\Gamma)$ is equivalent to maximizing the dominant characteristic root $r_0(\Gamma)$ as a function of $\Gamma$ (cf. [3]).

This theorem leads to two types of arguments: Let the maximized $\psi_r[u](\Gamma)$ be given by

$$\tilde{\psi}_r(\Gamma) := \int_A dy \, \sup_{u \in \mathbb{U}} \left\{ \int_0^\alpha da \, \exp\{-ra\} F(a, y, u, \Gamma)(U(a, 0, u, \Gamma)v)(y) \right\}. \tag{27}$$

One argument is related to the $r$ selection theory that maximizes the dominant characteristic root when we choose the condition

$$\tilde{\psi}_r(\Gamma)\Big|_{r=\tilde{r}, \Gamma=\mathbf{o}} = 1. \tag{28}$$

Because $\Gamma$ represents the strength of the density effects, $\Gamma = \mathbf{o}$ indicates the adaptive strategy that will satisfy the selection of $r$. The second argument represents the conditions in $K$ selection:

$$\psi_r[u](\Gamma)\Big|_{r=0, \Gamma=\tilde{\Gamma}} \leq \tilde{\psi}_r(\Gamma)\Big|_{r=0, \Gamma=\tilde{\Gamma}} = 1,$$

because the adaptive strategy in a stationary population is believed to be uninvaded by any strategy. $\psi_0[u](\Gamma)$ is essentially the basic reproductive number, and, therefore,

$$\tilde{\psi}_r(\Gamma)\Big|_{r=0, \Gamma=\tilde{\Gamma}} = 1 \tag{29}$$

is necessary and sufficient for the adaptive strategy in $K$ selection ($K$ strategy). $\tilde{\Gamma}$ must satisfy several additional conditions, such as existence, uniqueness, and stability. The details of these additional conditions can be determined in Text A in S1 File. Although the $r$ strategy cannot serve to conserve the exponential growth of the population in nature, it is believed to be the case that the $r$ strategy matches the $K$ strategy. In this case, the $r$ strategy comprising precocity and prolificacy becomes a candidate for the adaptive strategy even in a stationary population. For example, there is a mathematical model in which intraspecific competition does not influence the control of foraging resources [3]. If $v(y) = \delta^d(x - y)$, our method unifies the $r/K$ strategies via the characteristic function in Eq (27), which is matched with the consequence in the references mentioned previously.

$\Gamma$ is adjusted to

$$\Gamma = \Gamma^\dagger \in \left\{ \Gamma \in \mathbb{R}_+^{d'} \Big| r_0(\Gamma) = 0 \right\}$$

assuming that each element is positive for all $\ell'$:

$$0 < \Gamma^{\dagger\ell'} < \infty.$$

Then, a population density $P^\dagger(a, y)$ generating $\Gamma^\dagger$ exists and satisfies

$$\frac{\partial}{\partial a} P^\dagger(a, y) = -H(a, y, v, \Gamma^\dagger) P^\dagger(a, y)$$

$$P^\dagger(0, y) = \beta^\dagger v(y)$$

$$\beta^\dagger : = \langle F, P^\dagger \rangle \Big|_{\Gamma = \Gamma^\dagger} = \text{const.} \Leftrightarrow \psi_r(\Gamma) \Big|_{r=0, \Gamma=\Gamma^\dagger} = 1.$$

Therefore, $\Gamma^\dagger$ can provide a saturated population under nonlinear population dynamics.

Let us consider the maximized function

$$\tilde{v}_r(a, y, \Gamma) := \sup_{u \in \mathbb{U}} \{v_r[u](a, y, \Gamma)\} \tag{30}$$

$$v_r[u](a, y, \Gamma) = \int_a^\alpha d\tau \ \exp\{-r(\tau - a)\} U^*[u](a, \tau, \Gamma) v(0) v F(\tau, u_\tau, \Gamma, \cdot). \tag{31}$$

By applying the stochastic interpretation to Eq (31), Eq (30) can be rewritten as the statistics of a diffusion process $X_\tau = (X_\tau^j)_{1 \le j \le d}$ as

$$\tilde{v}_r(a, y, \Gamma) =$$

$$\sup_{u \in \mathbb{U}} \left\{ \tilde{\psi}_r(\Gamma) \mathbb{E}_y \left[ \int_a^\alpha d\tau \ F(\tau, X_\tau, u_\tau, \Gamma) \exp\left\{ -\int_a^\tau ds \ (\mu(s, X_s, u_s, \Gamma) + r) \right\} \right] \right\}, \tag{32}$$

where $\mathbb{E}_y[\cdot]$ denotes the expectation of the probability measure of $X_\tau$ in $X_a = y$. This representation is called the Feynman–Kac formula, and it is well known in stochastic analysis [19]. Eq (32) is called the value function in the control theory [21]. The diffusion process $X_\tau = (X_\tau^j)_{1 \le j \le d}$ satisfies the following stochastic differential equation (SDE):

$$X_\tau^j = y^j + \int_a^\tau ds \ g_j(s, X_s, u_s, \Gamma) + \sum_{\ell=1}^N \int_a^\tau \sigma_{j\ell}(s, X_s, u_s, \Gamma) dB_s^\ell \quad a \le \tau \le \alpha$$

$$X_a^j = y^j$$

The SDE is given by Eq (1) parameterized by $u$ and $\Gamma$, and it can describe the growth process of each state from age $a$ to $u$ in both trivial ($\Gamma = 0$) and nontrivial ($\Gamma = \Gamma^\dagger$) equilibrium points.

Thus, $v_r[u](a, y, \Gamma)$, the solution of the Dirichlet problem

$$\left[\frac{\partial}{\partial a} - H^*(a, y, u, \Gamma) - r\right] v_r[u](a, y, \Gamma) + \tilde{v}_r(\Gamma) F(a, y, u, \Gamma) = 0,$$

provides a statistical representation of the corresponding diffusion process called the Feynman–Kac formula [19, 22]. The adjoint Hamiltonian is given by

$$H^*(a, y, v, \Gamma) =$$

$$-\sum_{j=1}^d g_j(a, y, u, \Gamma) \frac{\partial}{\partial y^j} - \frac{1}{2} \sum_{\substack{j=1 \\ j'=1}}^d S_{jj'}(a, y, u, \Gamma) \frac{\partial^2}{\partial y^j \partial y^{j'}} + \mu(a, y, u, \Gamma)$$

The stochastic interpretation is appropriate for describing the adaptive life history and corresponding population dynamics for the following two reasons. (1) To reveal that the fittest dynamics are generated by the optimally controlled life history of individuals. (2) To derive the main equation in this study from the central principle of optimality efficiently.

According to the optimal control theory, adaptive strategies must follow a basic property called Bellman's principle (or the principle of optimality):

"an optimal strategy has the property that whatever the initial state and initial control are, the remaining control must constitute an optimal strategy with regard to the state resulting from the first strategy" [23].

The following relationship is derived based on this principle:

$$
\tilde{v}_r(a_0, y, \Gamma) =
$$
$$
\sup_{u \in \mathbb{U}} \left\{ \mathbb{E}_y \left[ \tilde{v}_r(a, X_a, \Gamma) \exp \left\{ - \int_{a_0}^a ds \ (\mu(s, X_s, u_s, \Gamma) + r) \right\} \right. \right.
$$
$$
\left. \left. + \tilde{\psi}_r(\Gamma) \int_{a_0}^a ds \ F(s, X_s, u_s, \Gamma) \exp \left\{ - \int_{a_0}^s d\tau \ (\mu(\tau, X_\tau, u_\tau, \Gamma) + r) \right\} \right] \right\},
$$
(33)

where $0 \leq a_0 \leq a \leq \alpha$. This relationship implies that the adaptive control from $a_0$ to $a$ in the terminal condition $\tilde{v}_r(a, y, \Gamma)$ is consistent with the control of this function from $a_0$ to $\alpha$, and it leads to

$$
\frac{\partial}{\partial a} \tilde{v}_r(a, y, \Gamma) - \inf_{u \in \mathbb{U}} \left\{ [H^*(a, y, u, \Gamma) + r] \tilde{v}_r(a, y, \Gamma) - \tilde{\psi}_r(\Gamma) F(a, y, u, \Gamma) \right\} = 0
$$
$$
\tilde{v}_r(\alpha, y, \Gamma) = 0
$$
(34)

$$
\tilde{\psi}_r(\Gamma) = \int_A dy \ \tilde{v}_r(0, y, \Gamma) v(y),
$$
(35)

(see Text B in S1 File). This equation is significant in control theory and is called the Hamilton–Jacobi–Bellman (HJB) equation. From the basic theorem of the adaptive life schedule, the adaptive strategy $\tilde{u}_r(a, y)\big|_{y=X_a}$ in $r$ selection ($r$ strategy) is obtained using Eq (28), and the $K$ strategy $\tilde{u}_K(a, y)\big|_{y=X_a}$ is given by Eq (29).

Based on Eqs (28) and (29), Eq (34) is simplified as

$$
\frac{\partial}{\partial a} \tilde{v}_r(a, y, \Gamma) - \inf_{u \in \mathbb{U}} \left\{ [H^*(a, y, u, \Gamma) + r] \tilde{v}_r(a, y, \Gamma) - F(a, y, u, \Gamma) \right\} = 0,
$$
$$
\tilde{v}_r(\alpha, y, \Gamma) = 0
$$
$$
\tilde{\psi}_r(\Gamma) = 1
$$
(36)

Thus, we obtain an equation for which the adaptive strategy is satisfied in a constant environment. Eqs (28), (29) and (36) contain and are more general than the result of [3] because they account for reproductive controls. Moreover, these equations reveal that adaptive control depends on the state distribution of the neonatal population $v(y)$ via $\tilde{r}_0$ or $\tilde{\Gamma}$. Accordingly, the equation above indicates that individual life histories evolve to maximize the reproductive value function (Eq (32) at age zero) in a constant environment.

## External stochasticity and perturbation method

The previous sections revealed a parameter that maximizes the adaptive life history in a constant environment. This section presents the population dynamics behavior under a simple stochastic environment.

Although there are several assumptions and candidates for statistical noise as external stochasticity, we simplify environmental stochasticity as white noise parameterized by $a$ and $y$

$\mathscr{W}_t(a, y)$.

$$\mathscr{W}_t(a, y) = \frac{\partial}{\partial t} \mathcal{B}_t(a, y) \tag{37}$$

$$\mathbb{E}^{\text{ext}}[\mathscr{W}_t(a, y)] = 0 \tag{38}$$

$$\mathbb{E}^{\text{ext}}[\mathscr{W}_t(a, y)\mathscr{W}_t(a', y')] = \delta(a - a')\delta^d(y - y'), \tag{39}$$

for all $t > 0$. $\mathcal{B}_t(a, y)$ denotes the Brownian motion parameterized by $a$ and $y$. Consider that a population vector under external stochasticity $P_t^\varepsilon(a, y)$ follows the stochastic partial differential equation

$$\left(\frac{\partial}{\partial t} + \frac{\partial}{\partial a}\right) P_t^\varepsilon(a, y) = -(H(a, y) - \varepsilon\mathscr{W}_t(a, y))P_t^\varepsilon(a, y), \tag{40}$$

where $\varepsilon$ denotes a sufficiently small positive constant that represents the strength of external stochasticity. Because it is difficult to compute a strict value of an LLGR involving external stochasticity, we apply a perturbation method to $\varepsilon$ to calculate its approximate value, such that

$$\bar{r}(\varepsilon) = r_0 + \varepsilon D_1 + \varepsilon^2 D_2 + \cdots.$$

**Second-order approximation of long-term logarithmic growth rate.** We introduced the derivation of the second-order approximation of LLGR in Eq (40). The population Hamiltonian vector, Hamiltonian, and noise functions are simplified to avoid computational complexity as

$$\begin{aligned} P_t^\varepsilon &= P_t^\varepsilon(a, y) \\ \mathscr{W}_t &= \mathscr{W}_t(a, y). \end{aligned}$$

Let us consider the following variation of the constants formula:

$$P_t^\varepsilon = T(t)\varphi + \varepsilon \int_0^t dt' \ T(t - t')\mathscr{W}_{t'}P_{t'}^\varepsilon. \tag{41}$$

The semigroup $T(t)$ is defined by Eq (21). With Eq (41) and Ito's formula for the multiple stochastic integral [24], a perturbation of the population vector is found by computing iteratively.

$$P_t^\varepsilon = \sum_{m=0}^\infty \varepsilon^m Q_m(t)\varphi$$

$$Q_0(t)\varphi := T(t)\varphi = \sum_{k=0}^n \langle v_k, \varphi \rangle \exp\{r_k t\} w_k + O(\exp\{(\Re r_k - \epsilon)t\}) \tag{42}$$

$$Q_m(t)\varphi :=$$

$$\int_0^t \cdots \int_0^{t^{(m-1)}} dt' dt'' \cdots dt^{(m)} \ T(t - t')\mathscr{W}_{t'} T(t' - t'')\mathscr{W}_{t''} T(t'' - t''') \cdots \mathscr{W}_{t^{(m)}} T(t^{(m)})\varphi$$

Introducing a new operation symbol

$$\|f\| = \int_0^\alpha da \int_A dy \ f(a, y),$$

each $Q_m(t)\varphi$ is deformed as

$$
\begin{aligned}
Q_m(t)\varphi \quad &= \sum_{k_1,\cdots,k_m}^n \exp\{r_{k_m}t\}w_{k_m} \\
&\times \int_0^t \cdots \int_0^{t^{(m-1)}} dt'dt'' \cdots dt^{(m)} \prod_{\ell=1}^{m-1} \|v_{k_{m-\ell}} \mathscr{W}_{t^{(m-\ell)}} w_{k_{m-\ell+1}}\| \, \|v_{k_m} \mathscr{W}_{t^m}\varphi\| \\
&= \sum_{k_1,\cdots,k_m}^n \exp\{r_{k_m}t\}w_{k_m} \int_0^t \cdots \int_0^{t^{(m-1)}} \prod_{\ell=1}^{m-1} \|v_{k_{m-\ell}} d\mathcal{B}_{t^{(m-\ell)}} w_{k_{m-\ell+1}}\| \, \|v_{k_m} d\mathcal{B}_{t^m}\varphi\| \\
&= \frac{t^{\frac{m}{2}}}{m!} \sum_{k_1,\cdots,k_m}^n \exp\{r_{k_m}t\}w_{k_m} \Big\|h_m\Big(\frac{\mathcal{B}_t}{\sqrt{t}}\Big)v_{k_m}\varphi \prod_{\ell=1}^{m-1} v_{k_\ell} w_{k_\ell}\Big\|,
\end{aligned}
$$

(43)

where $h_m(x)$ denotes the Hermite polynomial

$$
h_m(x) := (-1)^m \exp\Big\{\frac{x^2}{2}\Big\} \frac{d^m}{dx^m} \exp\Big\{-\frac{x^2}{2}\Big\} \quad m = 0, 1, 2, \cdots.
$$

The last row in Eq (43) is derived from the following formula [24]:

$$
m! \int_0^t \int_0^{t'} \cdots \int_0^{t^{(m-1)}} dB_{t'} dB_{t''} \cdots dB_{t^{(m)}} = t^{\frac{m}{2}} h_m\Big(\frac{B_t}{\sqrt{t}}\Big) \quad t \geq t' \geq \cdots \geq t^{(m)}.
$$

The arbitrary constant of the adjoint eigenfunction is set as

$$
\int_A dx \, v_k(0,x)v(x) = \langle v_k, w_k \rangle^{-1} \quad k = 0, 1, 2, \cdots.
$$

If the population vectors in the presence and absence of external stochasticity are close to each other, $P_t^\varepsilon \approx P_t^0 = P_t$ ($\varepsilon \ll 1$), the perturbation expressed in Eq (43) provides an accurate approximation. With this assumption, a $\varepsilon$-specific mean LLGR $\bar{r}_E(\varepsilon)$ is represented by substituting Eq (43) into Eq (23) such that

$$
\bar{r}_E(\varepsilon) = \lim_{t\uparrow\infty} \frac{1}{t} \mathbb{E}^{ext}\big[\ln\|P_t^\varepsilon\|_X\big] = \lim_{t\uparrow\infty} \frac{1}{t} \mathbb{E}^{ext}\Big[\ln\Big\{\sum_{m=0}^\infty \varepsilon^m \langle v_0, Q_m(t)w_0\rangle\Big\}\Big].
$$

(44)

For simplicity, suppose that the initial population is the eigenfunction corresponding to the 0–zeroth characteristic root.

By expanding Eq (44) into a Taylor series at $\varepsilon = 0$, the growth rate becomes

$$
\begin{aligned}
\bar{r}_E(\varepsilon) = r_0 \quad &+ \lim_{t\uparrow\infty} \frac{\varepsilon}{t} \mathbb{E}^{ext}\Big[\frac{\langle v_0, Q_1(t)w_0\rangle}{\langle v_0, Q_0(t)w_0\rangle}\Big] \\
&+ \lim_{t\uparrow\infty} \frac{\varepsilon^2}{t} \mathbb{E}^{ext}\Big[\frac{\langle v_0, Q_2(t)w_0\rangle}{\langle v_0, Q_0(t)w_0\rangle} - \frac{1}{2}\Big(\frac{\langle v_0, Q_1(t)w_0\rangle}{\langle v_0^*, Q_0(t)w_0\rangle}\Big)^2\Big] \\
&+ O(\varepsilon^3).
\end{aligned}
$$

(45)

Let us consider the mean growth rate in an environment comprising sufficiently small disturbances such that the third- (or higher-) order terms in $\varepsilon$ can be truncated. The second term on the right-hand side is zero in the mean growth rate because of the statistical property of the fluctuation term (cf. Eq (38)). Accordingly, the key point is the estimation of the second-order

term in Eq (45). One of the pieces composing the second-order term is computed as

$$\langle v_0, Q_2(t)w_0 \rangle = \exp\{r_0 t\} \frac{t}{2!} \sum_k \|h_2\left(\frac{\mathcal{B}_t}{\sqrt{t}}\right) v_0 w_0 v_k w_k\|. \tag{46}$$

Hermite polynomials are orthogonal with respect to Gaussian measure

$$\frac{1}{\sqrt{2\pi}} \int_{-\infty}^{\infty} dx \ h_m(x) h_{m'}(x) \exp\left\{-\frac{x^2}{2}\right\} = m!\delta_{mm'},$$

i.e., the term becomes statistically zero.

$$\mathbb{E}^{\text{ext}}[\langle v_0, Q_2(t)w_0 \rangle] = 0. \tag{47}$$

Similarly, the other component of the third term in Eq (45) is computed as

$$\frac{1}{2}\lim_{t\uparrow\infty}\frac{1}{t}\mathbb{E}^{\text{ext}}\left[\left(\frac{\langle v_0, Q_1(t)w_0 \rangle}{\langle v_0, Q_0(t)w_0 \rangle}\right)^2\right] = \frac{1}{2}\lim_{t\uparrow\infty}\frac{1}{t}\int_0^t dt' \ \|(v_0 w_0)^2\|$$

$$= \frac{1}{2}\|(v_0 w_0)^2\|. \tag{48}$$

After combining the components (Eqs (47) and (48)), the second-order approximation of the LLGR becomes

$$\bar{r}_E(\varepsilon) \approx r_0 - \underbrace{\frac{\varepsilon^2}{2}\|(v_0 w_0)^2\|}_{\text{deviation term}}. \tag{49}$$

This approximation is similar to the Tuljapurkar approximation [7]; however, it differs in several aspects. For instance, the deviation term corresponding to the original Tuljapurkar approximation is described by a sensitivity matrix. In this continuous version, statistics concerning the diffusion process $X_a$ account for the term. One important point is that the second term on the right-hand side of the equation above incorporates eigenfunctions. As described previously, the adjoint eigenfunction serves as an objective function to determine the adaptive strategy. This characteristic suggests that an adaptive species in a variable environment does not always maximize identical functions in a constant environment. That is, we may find another adaptive strategy $u^*$ as

$$\bar{r}_E[\tilde{u}_r] \approx \tilde{r} - \frac{\varepsilon^2}{2}\|(\tilde{v}_0 \tilde{w}_0)^2\| \le \bar{r}_E[u^*],$$

where the arbitrary constant is set to

$$\int_A d\xi \ \tilde{v}_0(0,\xi)v(\xi) = \langle \tilde{v}_0, \tilde{w}_0 \rangle^{-1}.$$

## Results

### Specific model for twofold stochasticity

The previous section revealed that the effect of external stochasticity on population growth is represented by the eigenfunctions corresponding to the dominant characteristic root in the mean environment. We use a specific mathematical model that is analytically solvable to examine the contribution of internal stochasticity to external stochasticity.

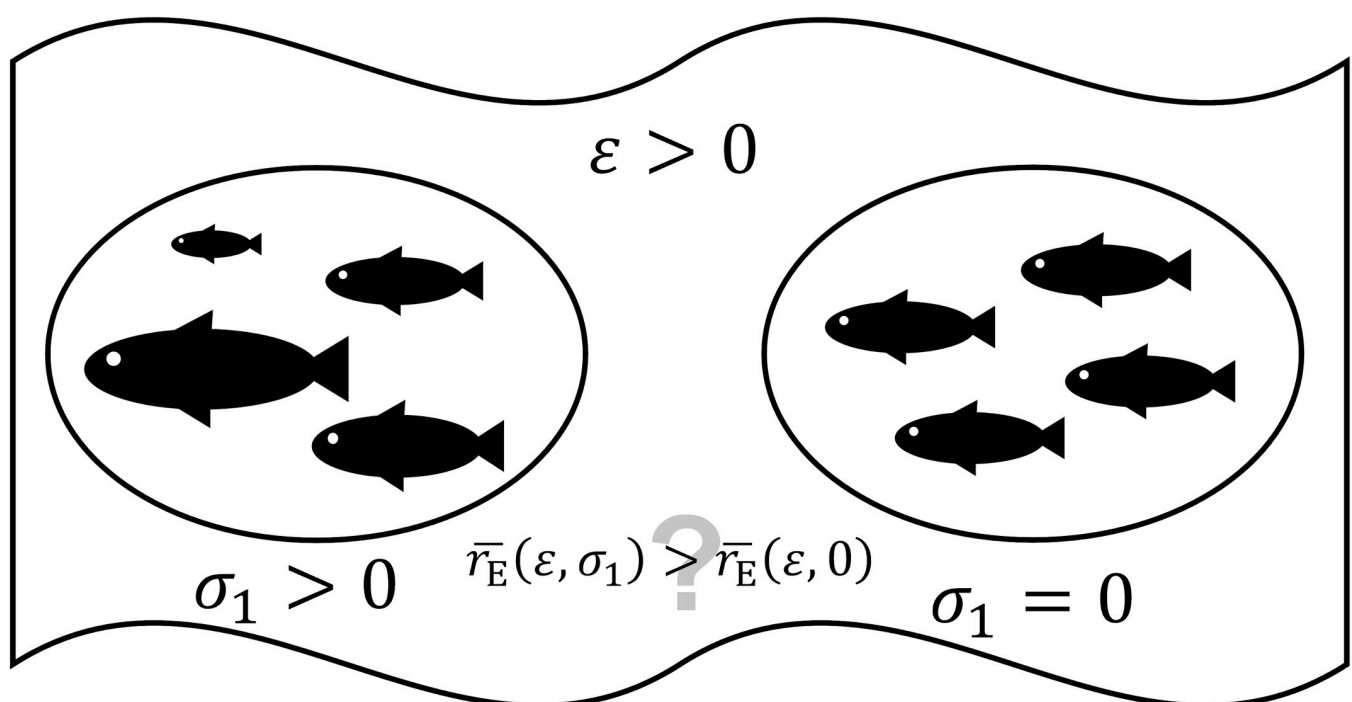

**Fig 1. Variation in individual size growth in a variable environment.**

Let us consider the role of internal stochasticity in external stochasticity. We construct a mathematical model that compares the LLGRs on a group of inhomogeneous growth rates with those of a group of homogeneous growth rates in a variable environment (cf. Fig 1). This figure illustrates the concept of a simple model. This model verifies whether the variance in size growth $\sigma_1$ increases with the LLGR $\bar{r}_E(\sigma_1)$ for positive values of $\varepsilon$.

The model aims to estimate the existence of the adaptive control of internal stochasticity against external stochasticity. As indicated in the aforementioned analyses of matrix models based on empirical data, if organisms control their growth rate statistics, there exists an adaptive strength of heterogeneity.

When $X_a \in \mathbb{R}_+$ is the size at age $a \in [0, \infty)$, as an effect of internal stochasticity, we assume that the heterogeneity of the individual size growth rate is

$$
\begin{aligned}
dX_a &= b_1 X_a da + \sigma_1 X_a dB_a^1 \ (b_1, \sigma_1 > 0) \\
X_0 &= x
\end{aligned}
\tag{50}
$$

Heterogeneity is generated by the fluctuation in the second term on the right-hand side of Eq (50). The SDE provides a geometric Brownian motion that grows exponentially with the fluctuation. Suppose that mortality is constant. Then,

$$
\mu(a, y) \equiv \mu_0 = const.
\tag{51}
$$

Fertility is assumed to be an allometric function in size

$$
F(a, y) = f_0 y^\rho \ \ 0 < \rho < 1.
\tag{52}
$$

This life history generates the following Hamiltonian and adjoint Hamiltonian.

$$H(a, y)\phi(y) = \mathcal{H}_y\phi(y) = \frac{\partial}{\partial y}(b_1 y \phi(y)) - \frac{1}{2}\frac{\partial^2}{\partial y^2}\left(\sigma_1^2 y^2 \phi(y)\right) + \mu_0 \phi(y) \tag{53}$$

$$H^*(a, y)\phi(y) = \mathcal{H}_y^*\phi(y) = -b_1 y \frac{\partial}{\partial y}\phi(y) - \frac{1}{2}\sigma_1^2 y^2 \frac{\partial^2}{\partial y^2}\phi(y) + \mu_0 \phi(y), \tag{54}$$

respectively. Assuming all neonates have identical state $x$

$$v(y) = \delta(x - y),$$

eigenfunction $w_r$ satisfies

$$\left(\frac{\partial}{\partial a} + \mathcal{H}_y + r\right) w_r(a, y) = 0, \quad \varphi_r(0, y) = \delta(x - y). \tag{55}$$

Substituting an ansatz

$$w_r(a, y) = \exp\left\{-(\mu_0 + r)a\right\}\phi(a, y)$$

into Eq (55), the equation is converted into a Fokker–Planck equation

$$\left(\frac{\partial}{\partial a} + \mathcal{H}_y - \mu_0\right)\phi(a, y) = 0, \quad \lim_{a\downarrow 0}\phi(a, y) = \delta(x - y),$$

which gives the probability density function of the geometric Brownian motion in Eq (50). The probability density function is then given by the logarithmic normal distribution

$$\phi(a, y) = \frac{1}{y\sqrt{2\pi\sigma_1^2 a}}\exp\left\{-\frac{\left(\ln\frac{y}{x} - \left(b_1 - \frac{\sigma_1^2}{2}\right)a\right)^2}{2\sigma_0^2 a}\right\}.$$

Therefore, the eigenfunction is

$$w_r(a, y) = \frac{1}{y\sqrt{2\pi\sigma_1^2 a}}\exp\left\{-\frac{\left(\ln\frac{y}{x} - \left(b_1 - \frac{\sigma_1^2}{2}\right)a\right)^2}{2\sigma_1^2 a} - (\mu_0 + r)a\right\}. \tag{56}$$

Because the size growth rate follows an age-homogeneous Markovian process (Eq (50)), this adjoint function does not depend on age.

$$\begin{aligned}
v_r(a, y) &= v_r(0, x)\lim_{\alpha\uparrow\infty}\mathbb{E}_y\left[\int_a^\alpha ds\, f_0 X_s^\rho \exp\left\{-(r + \mu_0)(s - a)\right\}\right] \\
&= v_r(0, x)\lim_{\alpha\uparrow\infty}\int_a^\alpha ds\, \exp\left\{-(r + \mu_0)(s - a)\right\}\mathbb{E}_y[f_0 X_{s-a}^\rho] \\
&= v_r(0, x)\lim_{\alpha\uparrow\infty}\int_0^{\alpha-a} ds'\, \exp\left\{-(r + \mu_0)s'\right\}\mathbb{E}_y[f_0 X_{s'}^\rho] \\
&= v_r(x)\int_0^\infty ds'\, \exp\left\{-(r + \mu_0)s'\right\}\mathbb{E}_y[f_0 X_{s'}^\rho] \\
&= v_r(y).
\end{aligned} \tag{57}$$

Therefore, we have

$$v_r(a, y) = v_r(y). \tag{58}$$

Then, the adjoint eigenfunction follows the adjoint equation

$$-(\mathcal{H}_y^* + r)v_r(y) + v_r(x)f_0 y^\rho = 0. \tag{59}$$

This equation is explicitly solvable using the following ansatz:

$$v_r(y) = C_r y^\rho \quad C \neq 0,$$

which gives

$$v_r(y) = C_r y^\rho = \frac{v_r(x)f_0 y^\rho}{-b_1\rho + \dfrac{1}{2}\sigma_1^2\rho(1-\rho) + \mu_0 + r}. \tag{60}$$

Because the function above can compose the characteristic equation

$$\psi_r\big|_{r=r_0} = v_r(x)\big|_{r=r_0} = 1, \tag{61}$$

the dominant characteristic root is computed as

$$r_0 = \rho\left(b_1 - \frac{1}{2}\sigma_1^2(1-\rho)\right) + f_0 x^\rho - \mu_0. \tag{62}$$

By the definition of $0 < \rho < 1$, the characteristic root indicates that internal stochasticity has a negative effect on population growth in a constant environment

$$\frac{\partial r_0}{\partial \sigma} < 0.$$

That is, it is nonadaptive for species to have heterogeneity under the condition $0 < \rho < 1$.

Substituting the dominant characteristic root Eq (62) into Eq (60), the functional becomes

$$\langle v_0, w_0 \rangle = v_r(x)\frac{d}{dr}\psi_r\Big|_{r=r_0} = \frac{1}{f_0 x^\rho}.$$

Hence, the arbitrary constant is determined to be

$$v_0(x) = \langle v_0, w_0 \rangle^{-1} = f_0 x^\rho,$$

and the adjoint eigenfunction corresponding to the dominant characteristic root is

$$v_0(y) = f_0 y^\rho. \tag{63}$$

In this case, the adjoint eigenfunction corresponding to the dominant root matches the fertility function.

Combining the previous steps, the deviation term is given by

$$
\begin{aligned}
\frac{\varepsilon^2}{2}\left\|(v_0 w_0)^2\right\| &= \frac{\varepsilon^2}{2}\int_0^\infty\int_{\mathbb{R}_+} da\, dy\, \frac{f_0^2 y^{-2(1-\rho)}}{2\pi\sigma_1^2 a}\\
&\times\left.\exp\left\{-\frac{2\left(\ln\frac{y}{x}-\left(b_1-\frac{\sigma_1^2}{2}\right)a\right)^2}{2\sigma_1^2 a}-2(\mu_0+r)a\right\}\right|_{r=r_0}\\
&= \frac{\varepsilon^2 f_0^2 x^{2\rho-1}}{4\sigma_1\sqrt{b_1\left(\rho(2-\sqrt{2})+\frac{\sqrt{2}}{2}\right)+\frac{1}{2}\sigma_1^2\left(\sqrt{2}\rho+\frac{\sqrt{2}-1}{2}\right)+f_0 x^\rho}}.
\end{aligned}
\tag{64}
$$

This deviation term diverges to infinity in the absence of internal stochasticity

$$
\sigma_1\to 0,\quad \frac{\varepsilon^2}{2}\left\|(v_0 w_0)^2\right\|\to\infty.
$$

Thus, it is reasonable to consider the effect of higher orders of $\varepsilon$ on this divergence. However, this consequence suggests the significance of heterogeneity in the persistence of species in minimally variable environments. This property contrasts with the effect of internal stochasticity on the dominant characteristic root (Eq (62)). Substituting Eqs (62) and (64) into Eq (49), the LLGR approximates

$$
\begin{aligned}
\bar{r}_E(\varepsilon) &\approx \rho\left(b_1-\frac{1}{2}\sigma_1^2(1-\rho)\right)+f_0 x^\rho-\mu_0\\
&-\frac{\varepsilon^2 f_0^2 x^{2\rho-1}}{4\sigma_1\sqrt{b_1\left(\rho(2-\sqrt{2})+\frac{\sqrt{2}}{2}\right)+\frac{1}{2}\sigma_1^2\left(\sqrt{2}\rho+\frac{\sqrt{2}-1}{2}\right)+f_0 x^\rho}}.
\end{aligned}
\tag{65}
$$

The LLGR represents a monotonically increasing function with respect to the mean size growth rate $b_1$,

$$
\frac{\partial\bar{r}_E(\varepsilon)}{\partial b_1}\geq 0.
$$

This point may appear to be trivial, yet it is notable that the deviation term monotonically decreases in $b_1$. Further, rapid growth may reduce the risks inherent to variable environments.

The hHeterogeneity of the size growth rate reduces the mean dominant characteristic root and causes the risk of extinction from the variable environment. By computing Eq (65) in terms of $\sigma$ and $\varepsilon$, we find the adaptive heterogeneity of the size growth rate for each $\varepsilon$ (see Fig 2). This figure shows the existence of an adaptive value in $\sigma_1$. Each $\varepsilon$ representing the strength of external stochasticity has a unique adaptive value of $\sigma_1$ that maximizes the dominant characteristic root $r_0$; $\varepsilon$ increases with an adaptive value $\sigma_1$. This result suggests that species require greater heterogeneity in more variable environments. The parameters are $b_1 = 0.6$, $x = 0.01$, $\mu_0 = 0.1$, $f_0 = 1.0$, and $\rho = 0.4$.

Fig 2 illustrates that adaptive heterogeneity increases with environmental variability. The numerical analysis suggests that species evolve to yield heterogeneity in variable environments. This viewpoint corroborates conventional interpretations of the necessity for biodiversity.

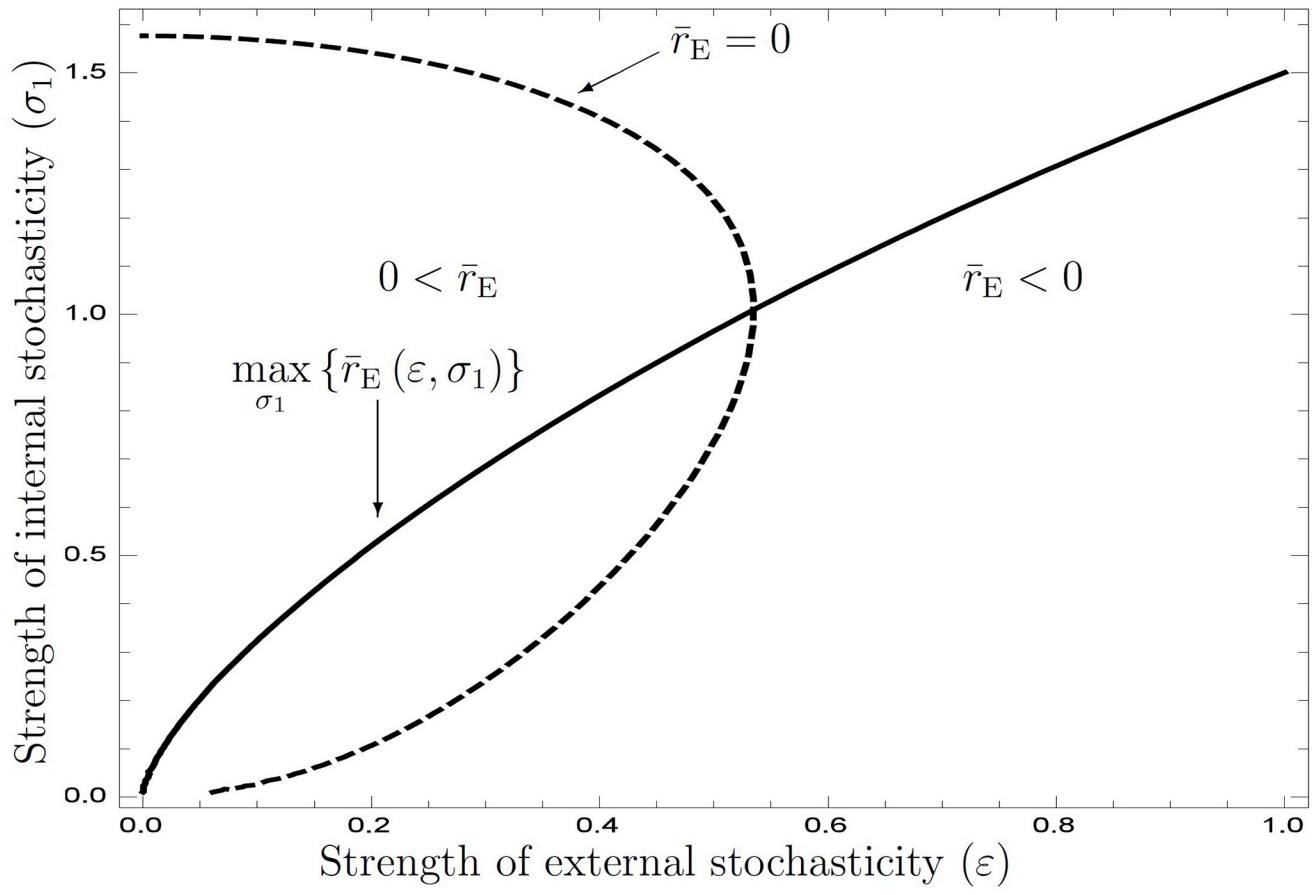

**Fig 2. Adaptive heterogeneity under two-fold stochasticity.**

### Adaptive resource utilization in external stochasticity

Based on Eqs (50)–(52), we consider a species utilizing different resources ($R_1$ and $R_2$). The specialist utilizing $R_1$ uses the size growth rate in Eq (50) and that of a specialist utilizing $R_2$ is

$$\begin{cases} dX_a^2 & = b_2 X_a^2 da + \sigma_2 X_a^2 dB_a^2 \\ X_0^2 & = x. \end{cases} \tag{66}$$

$B_a^1$ and $B_a^2$ are independent Brownian motions. Then, we assume that $b_1 \in \mathbb{R}_+ \geq b_2 \in \mathbb{R}$ ($b_2$ could be negative), $\sigma_1 > \sigma_2 \geq 0$, that is, choosing $R_1$ implies a higher risk and growth rate expectation than choosing $R_2$. Conversely, choosing $R_2$ under the same conditions confers another risk—that individuals have lower survival until they reach maturity than when choosing $R_1$ because of the slower average growth rate. Therefore, individuals should find their adaptive risk by hedging $\tilde{u}(a, X_a) \in [0, 1]$ in accordance with each population size under the following growth rate (cf. Fig 3).

$$\begin{cases} dX_a & = [b_1(1-u) + b_2 u]X_a da + [\sigma_1(1-u)dB_a^1 + \sigma_2 u dB_a^2]X_a \\ X_0 & = x, \end{cases} \tag{67}$$

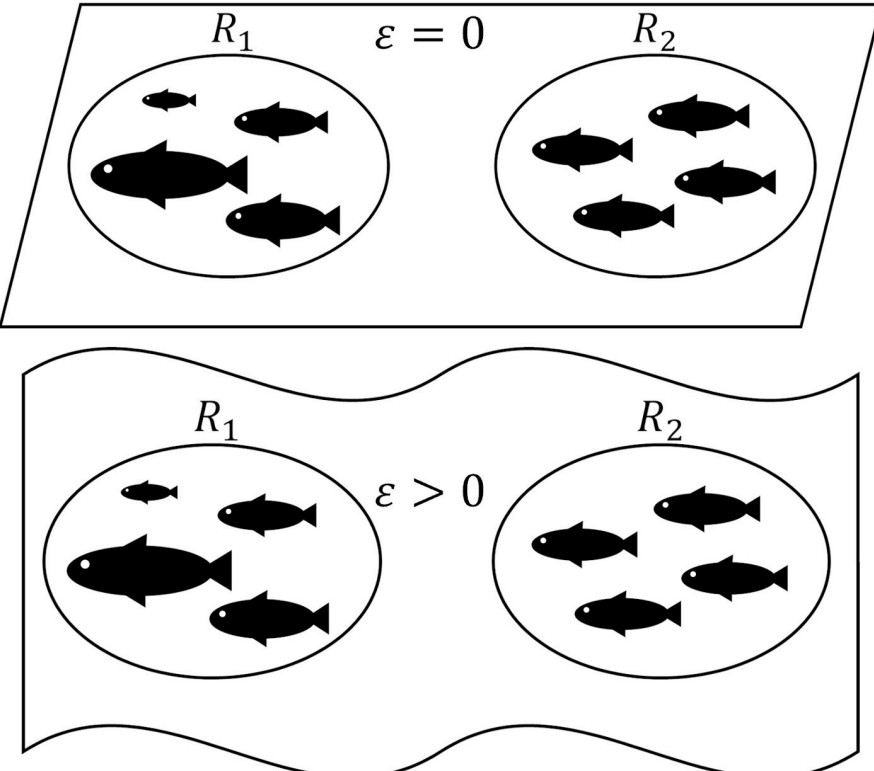

**Fig 3. Adaptive resource utilization in external stochasticity conditions.** This model is referred to as the two-resource utilization model and it generates the following adjoint Fokker–Planck Hamiltonian.

Fig 3 illustrates the concept of the adaptive resource utilization model. A resource $R_1$ provides the high size growth rate $b_1$ on average; however, the risk $\sigma_1$ is also high. Conversely, $R_2$ is low risk $\sigma_1 > \sigma_2$ and has a low size growth rate on average, i.e., $b_1 > b_2$. The species maximizes its LLGR by optimizing the utilization of both resources. Then, we verify that the existence of external stochasticity evolves different adaptive utilizations from that in a constant environment.

$$\mathcal{H}_y^*(u) =$$

$$-(b_1(1-u) + b_2 u)y\frac{d}{dy} - \frac{1}{2}\left(\sigma_1^2(1-u)^2 + \sigma_2^2 u^2\right)y^2\frac{d^2}{dy^2} + \mu_0. \tag{68}$$

In this model, finding the adaptive utilization is analogous to generatinge the optimal size growth curve with heterogeneity. This growth curve maximizes Eq (57) following our framework. Consequently, individuals adopting the adaptive allocation strategy compose the fittest species by maximizing the LLGR under $r$-selection.

Because the reproductive value is independent of age in this model, the value function (Eq (32)) also does not depend on age, such that

$$\tilde{v}_r(x) = \sup_{u\in[0,1]}\left\{\int_0^\infty ds' \ \exp\left\{-(r+\mu_0)s'\right\}\mathbb{E}_x[f_0 X_{s'}^\rho]\right\}.$$

From the Bellman's principle Eq (33), the value function has the following decomposition for

arbitrary age $a$:

$$
\begin{aligned}
\tilde{v}_r(x) = & \sup_{u \in [0,1]} \left\{ \int_0^a ds' \ \exp\{-(r+\mu_0)s'\} \mathbb{E}_x[f_0 X_{s'}^\rho] \right. \\
& \left. + \int_a^\infty ds' \ \exp\{-(r+\mu_0)s'\} \mathbb{E}_x[f_0 X_{s'}^\rho] \right\} \\
= & \sup_{u \in [0,1]} \left\{ \int_0^a ds' \ \exp\{-(r+\mu_0)s'\} \mathbb{E}_x[f_0 X_{s'}^\rho] + \mathbb{E}_x[\tilde{v}_r(X_a)] \exp\{-(r+\mu_0)a\} \right\}.
\end{aligned}
$$

The equation above is deformed as

$$
\begin{aligned}
0 = & \sup_{u \in [0,1]} \left\{ \int_0^a ds' \ \exp\{-(r+\mu_0)s'\} \mathbb{E}_x[f_0 X_{s'}^\rho] \right. \\
& \left. + \int_0^a d(\mathbb{E}_x[\tilde{v}_r(X_{s'})] \exp\{-(r+\mu_0)s'\}) \right\}.
\end{aligned}
\tag{69}
$$

Using the same process as that used for the derivation of the general HJB equation (see S.2.), adopt the Feynman–Kac formula [19, 21] into the equation above:

$$
d(\mathbb{E}_x[\tilde{v}_r(X_{s'})] \exp\{-(r+\mu_0)s'\}) = -ds \ \mathbb{E}_x[[\mathcal{H}_{X_s}^*(u) + r]\tilde{v}_r(X_{s'})] \exp\{-(r+\mu_0)s'\}.
$$

Take the limit as $a$ tends to zero such that

$$
\begin{aligned}
0 = & \lim_{a \downarrow 0} \frac{1}{a} \sup_{u \in [0,1]} \left\{ \int_0^a ds' \ \exp\{-(r+\mu_0)s'\} \mathbb{E}_x[f_0 X_{s'}^\rho] \right. \\
& \left. - \int_0^a ds \ \mathbb{E}_x[[\mathcal{H}_{X_s}^*(u) + r]\tilde{v}_r(X_{s'})] \exp\{-(r+\mu_0)s'\} \right\}.
\end{aligned}
\tag{70}
$$

Then, we have

$$
- \inf_{u \in [0,1]} \{[\mathcal{H}_x^*(u) + r]\tilde{v}_r(x)\} + f_0 x^\rho = 0.
\tag{71}
$$

The equation above implies that the adaptive control should provide an extreme value:

$$
\frac{\partial}{\partial u} \left( \mathcal{H}_x^*(u)\tilde{v}_r(x) \right) \bigg|_{u=\tilde{u}_r} = 0
$$

for all $x$. This necessary condition leads to the following relationship between adaptive utilization and the adjoint function.

$$
\tilde{u}_r = \frac{\sigma_1^2}{\sigma_1^2 + \sigma_2^2} + \frac{(b_1 - b_2)\frac{\partial}{\partial x}\tilde{v}_r(x)}{(\sigma_1^2 + \sigma_2^2)y\frac{\partial^2}{\partial x^2}\tilde{v}_r(x)}.
\tag{72}
$$

Thus, the control is independent of age, which is called stationary control in control theory. Substituting the adaptive control condition into the adjoint Hamiltonian

$$
-[\mathcal{H}_x^*(\tilde{u}_r) + r]\tilde{v}_r(x) + f_0 x^\rho = 0,
$$

we can derive the adjoint eigenfunction of the adaptive life history from the same ansatz, as in

Eq (60).

$$\tilde{v}_r(x) = \frac{f_0 x^\rho}{r - \lambda}$$

$$\lambda = \begin{cases} \rho\left(b_1 - \frac{1}{2}\sigma_1^2(1-\rho)\right) + f_0 x^\rho - \mu_0 & \tilde{u}_r = 0 \\ \left(\frac{b_1\sigma_2^2 + b_2\sigma_1^2}{\sigma_1^2 + \sigma_2^2}\right)\rho - \frac{1}{2}\frac{\sigma_1^2\sigma_2^2\rho(1-\rho)}{\sigma_1^2 + \sigma_2^2} + \frac{1}{2}\frac{(b_1-b_2)^2\rho}{(\sigma_1^2+\sigma_2^2)(1-\rho)} + f_0 x^\rho - \mu_0 & 0 < \tilde{u}_r < 1 \\ \rho\left(b_2 - \frac{1}{2}\sigma_2^2(1-\rho)\right) + f_0 x^\rho - \mu_0 & \tilde{u}_r = 1 \end{cases}.$$

From Eq (72) and the function above, adaptive utilization is computed as

$$\tilde{u} = \max\left\{\frac{\sigma_1^2}{\sigma_1^2 + \sigma_2^2} - \frac{(b_1 - b_2)}{(\sigma_1^2 + \sigma_2^2)(1-\rho)}, 0\right\}, \tag{73}$$

which is identical to the strategy in [16], and it is known as constant value control. It indicates that $R_2$-specific utilizers do not evolve. Because adaptive utilization is constant in constant environments, finding another utilization constant $u^*$ that maximizes the LLGR in a variable environment implies that another adaptive utilization exists, even if the constant is not optimal control. Suppose that utilization is always constant, and that $v^*$ becomes the adaptive strategy for twofold stochasticity. The utilization constant specific LLGR $\bar{r}[u](\varepsilon)$ considers whether the variable environment selects a life history that favors heterogeneity as adaptive. Because the utilization rate does not depend on age or size, we can consider a specific LLGR with the following change of coefficients in Eq (65).

$$b_1 \rightarrow b_1(1 - u) + b_2 v, \quad \sigma_1 \rightarrow \sqrt{\sigma_1^2(1-u)^2 + \sigma_2^2 u^2}.$$

Solving $\bar{r}[u](\varepsilon)$ numerically, we find that a variable environment favors heterogeneity, as suggested in the previous section (Fig 4). This figure illustrates adaptive resource utilization with respect to $\rho$ for several values of $\varepsilon$. Although $\rho$ represents the scaling exponent of fertility that denotes a measure of risk aversion in a deterministic environment, the adaptive utilization of risk appetite in the presence of external stochasticity exists in the domains of greater and smaller values of $\rho$. The domain of the adaptive strategy utilizing both resources narrows as $\varepsilon$ increases. This consequence is linked to the relationship between the adaptive value of $\sigma_1$ and $\varepsilon$ in Fig 1. The parameters are $b_2 = 0.5$, $\sigma_1 = 0.8$, $\sigma_2 = 0.005$, and $\varepsilon = \{0, 0.1, 0.3, 0.6, 0.9\}$; the others are the same as in Fig 1.

The scaling exponent $\rho$ represents a risk appetite index in economics; small values favor risk aversion. Adaptive strategy Eq (73) represents an identical interpretation of the exponent to that in economics. However, under external stochasticity $\varepsilon \neq 0$, minimal internal stochasticity does not become adaptive for small values of $\rho$.

## Discussion

This study attempted to construct a systematization of the optimal life schedule problem and population dynamics using the eigenfunction expansion of a structured population model. Our perturbation method was inspired by Tuljapurkar's approximation; however, our model is based on mathematical models of life scheduling that contain internal stochasticity (e.g., SDE). This change provides a theoretical basis for the argument that species pose environmental variability. By applying the framework in this study to this argument, we found that the optimal parameters reduce the risk of external stochasticity and increase the LLGR. Further,

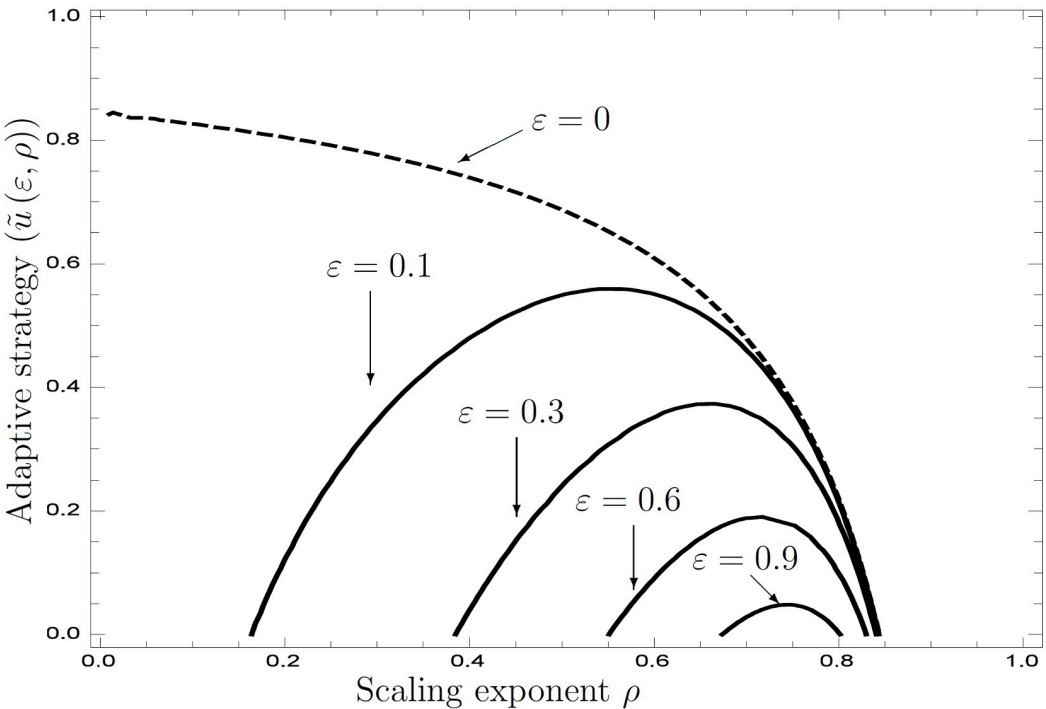

**Fig 4. Adaptive resource utilization under external stochasticity.**

the conventional adaptive life history in constant environments can be found using the HJB equation derived from the adjoint equation. If we regard the ESS in stable populations as an adaptive strategy, then using the HJB equation with additional parameters to represent the magnitude of the density effects on state in a stationary population can provide an adaptive life history under intraspecific competition.

The framework of this study helps reveal what evolution maximizes. In a constant environment, this framework extends the consequence in [3]: both adaptive strategies in the presence and absence of density effects maximize a common function. Further, although adaptive strategies under variable environments are less simple than those in constant environments are, this study shows that the effect of external stochasticity is closely related to eigenfunctions and Tuljapurkar's approximation. The second-order perturbation used in this study yielded trade-offs between the mean dominant characteristic root and the corresponding eigenfunctions via the LLGR. Given this relationship, the effect of internal stochasticity on the population growth may differ from its effect in a constant environment. Thus, as shown in the analysis of the specific model, the same adaptive strategies are not always used. External stochasticity needs to be treated as a different type of selection pressure than internal stochasticity and density effects. Therefore, deterministic models approximated by the averaged environment often overlook the essential adaptive strategies.

The specific model showed that deviation in the size growth rate buffers the reduction in LLGR caused by the variable environment. There is a trade-off between the decrease in the mean characteristic root and the buffering effect of a variable environment with internal stochasticity. This determines the adaptive heterogeneity of size growth, where the length is proportional to the magnitude of external stochasticity. These consequences support the premise that adaptive utilization prefers high-risk resources in a resource-utilization model in a variable environment. Compared with a constant environment, the domain of the allometric

exponent wherein species evolve with risk aversion under constant environments changes to a risk-taking strategy. Despite the small exponent value of a species, which indicates the brittleness of internal stochasticity in the deterministic LLGR, risky resources in a variable environment are selected. Therefore, these specific models appear to provide a theoretical basis for the conventional argument that individual heterogeneity is necessary for living in a variable environment. The mean rapid size growth in risky resources can be interpreted as having an advantage in terms of a small exponent value because fast growth statistically reduces the risk of external stochasticity. Considering that precocious species, such as mice, have short lifespans, this interpretation may be related to the short lifetime of organisms in variable environments [10, 11, 25]. However, in such a simple model, this interpretation requires careful consideration because the deviation term of the LLGR does not depend on mortality. Despite this simplification, our framework links empirical studies of evolution that pertain to life histories to various theoretical studies of structured population models.

The perturbation method in this study also avoids the mathematical complication of external stochasticity at the expense of biological correctness; incorporating these features remains an open problem. For instance, all cohorts should monotonically decrease with age; however, setting white noise as external stochasticity violates this rule. There is a limit to this study. Eq (40) can be interpreted as the fluctuation of mortality from external stochasticity; however, white noise neither correlates with each age-containing parameter nor ensures the positivity of mortality. This assumption is only for the sake of mathematical simplicity because it allows us to assume that external stochasticity alters the mean state growth rate and the fluctuation term from internal stochasticity or both. In this case, note the treatment of the derivatives in the noise function; these assumptions can obey the aforementioned biological rules because of the conservation law in the continuity equation. These noise functions are thought to complicate the problem and require considerable mathematical discussion. In addition, candidate stochastic processes are believed to vary such that the SDE can be defined by Ito's integral, Stratonovich's integral, and others [26, 27]. If we choose a noise function that does not have Markovian properties, the approximation of LLGR may not correspond with our results.

Disregarding the configuration method of each stochastic process, this study was conducted under the premise that all stochastic processes are assumed to be Markovian, which is an assumption that has been accepted by many ecologists. Structured population models have various versions including age-, size-, and stage-structured models; Tuljapurkar's approximation appears to work well in size- and stage-structured models that ignore cohort information. However, individuals and cohorts are essential elements when considering evolution in variable environments. Many empirical studies based on models that exclude cohort dynamics suggest a correlation between the transition rate and environmental variability [8]; however, they cannot clarify the strategy by which every individual"s life history reduces the risk of external stochasticity. On the other hand, these empirical studies suggest that the vital rate, which is important for adaptive strategy, is robust against environmental changes [9]. This suggestion imposes an important requirement on theoretical studies of life history evolution in a variable environment. Theoretical studies based on cohort dynamics should also consider this requirement. As mentioned in the Introduction, a twofold stochasticity perspective of cohort dynamics is necessary to understand the effect of stochasticity on life histories. Future transition matrix models must consider the age structure to understand how organisms oppose risk in a variable environment in their life history.

Thus far, theoretical research on the evolution of life history has been focused on an individual [28]. Within a lifespan, the strategy of maximizing the basic reproduction number is considered to be adaptive. The idea was the same in a variable environment [29]. The drawback of maximizing the basic reproduction number is that the generation time is not

considered. Therefore, it does not always match the maximization of $r_0$ and LLGR. In $r$–selection, the maximization of basic reproductive number may not be the optimal solution. Our framework overcomes that problem.

As shown in the analysis of the specific model in this study, internal and external stochasticity yield the diversity and extinction of organisms. Despite the simplicity of the assumptions of stochasticity, this study quantitatively demonstrates that heterogeneity decreases risks associated with a variable environment. Further, this result suggests that the existence of adaptive heterogeneity maximizes the population growth. Because many organisms are believed to have various adaptive strengths of traits, the diversity on an ecological scale may also occur. Eqs (28), (29) and (36) link the evolution of life history to population growth under internal stochasticity. Eq (49) connects the life history with the effect of external stochasticity via eigenfunctions. In $r$ selection, an adaptive strategy must optimize not only the basic reproductive number but also the generation time. An adaptive strategy in $K$ selection must generate density effects that prevent a stationary population from being invaded by other strategies. A previous study [3] posited that adaptive strategies in both $r$ and $K$ selection were identical via the common HJB equation, and this provides adjoint eigenfunctions.

On the other hand, the density effect from other states will depend on the current state. In addition, in terms of fertility, parental status is generally thought to affect the initial status of the offspring. In this study, these state dependences were ignored and assumed to be constant. Eliminating these assumptions will allow us to express more realistic intraspecific competition.

Consequently, this study shows that $r$ and $K$ selections and external stochasticity evolve different phenotypes; these selection pressures are independent of each other. In $r$ selection under a constant variable, our simple model shows that the heterogeneity should decrease because it decreases the expectation of the characteristic function. In the $K$ selection, the previous study demonstrated that the evolution of heterogeneity depended on how density effects operated in life history [3]. In $r$ selection under the variable environment, the homogeneity poses a high risk of extinction. The last one is caused by a trade-off between the mean growth rate and its variance in population dynamics. However, these results also suggest that the consequences of evolution in life history arise from optimizing a common factor, i.e., the reproductive value, in each habitat. To prove this, we must examine whether a life history adaptive strategy in more complicated environments (i.e., containing both density effects and external stochasticity) is explained by the framework developed in this study. The choice of the density effect and the definition of background noise (including non-Markovian) will generate numerous evolutions concerning heterogeneity in life history. Studies on these themes will find more sophisticated concepts of fitness. We hope that this research will be one of the cornerstones for future research.

## Supporting information

**S1 File. Text A, Stationary solutions and their local stability in K-selection**. Tex B, Derivation of the Hamilton-Jacobi-Bellman equation.
(ZIP)

## Acknowledgments

We are deeply grateful to Yuki Sugiyama for his helpful comments and advice on this study; Nobuhiko Fujii, Kensaku Kinjo, Youichi Enatsu, Kota Hattori, Tetsuya J. Kobayashi, and Akira Sakai for their mathematical advice; and Kumiko Oizumi, Kunihiro Aoki, Hiroko Oizumi, and Ryuichi Kaneko for their support and encouragement.

## Author Contributions

**Conceptualization:** Ryo Oizumi.

**Data curation:** Ryo Oizumi.

**Formal analysis:** Ryo Oizumi, Hisashi Inaba.

**Funding acquisition:** Ryo Oizumi.

**Investigation:** Ryo Oizumi.

**Methodology:** Ryo Oizumi.

**Project administration:** Ryo Oizumi.

**Resources:** Ryo Oizumi, Hisashi Inaba.

**Software:** Ryo Oizumi.

**Supervision:** Ryo Oizumi.

**Validation:** Ryo Oizumi.

**Visualization:** Ryo Oizumi.

**Writing – original draft:** Ryo Oizumi.

**Writing – review & editing:** Ryo Oizumi, Hisashi Inaba.

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
