## [Decision Letter · Decision Letter 0]

30 Jun 2021

PONE-D-21-12079

Evolution of heterogeneity under constant and variable environmets

PLOS ONE

Dear Dr. Oizumi,

Thank you for submitting your manuscript to PLOS ONE. After careful consideration, we feel that it has merit but does not fully meet PLOS ONE’s publication criteria as it currently stands. Therefore, we invite you to submit a revised version of the manuscript that addresses the points raised during the review process.

ACADEMIC EDITOR: The two reviewers have provided constructive and detailed comments. They both agreed that the work is interesting, relevant and would provide a good contribution. However, there are several aspects of the paper that need improvements, for which the reviewers have provided constructive suggestions. Please carefully consider them in the revision of your manuscript.

We look forward to receiving your revised manuscript.

Kind regards,

The Anh Han, Ph.D.

Academic Editor

PLOS ONE

Journal Requirements:

This research was supported by the Japan Society for the Promotion 587

of Science (JSPS) KAKENHI (Grant Number 20K14368) and the Ministry of Health, 588

Labour and Welfare under (Grant number 20AA2007). 

NO

4. We note that Figures 1 and 3 in your submission contain copyrighted images. All PLOS content is published under the Creative Commons Attribution License (CC BY 4.0), which means that the manuscript, images, and Supporting Information files will be freely available online, and any third party is permitted to access, download, copy, distribute, and use these materials in any way, even commercially, with proper attribution. For more information, see our copyright guidelines: http://journals.plos.org/plosone/s/licenses-and-copyright.

a. You may seek permission from the original copyright holder of Figures 1 and 3 to publish the content specifically under the CC BY 4.0 license. 

Additional Editor Comments:

The two reviewers have provided constructive and detailed comments. They both agreed that the work is interesting, relevant and would provide a good contribution. However, there are several aspects of the paper that need improvements, for which the reviewers have provided constructive suggestions. Please carefully consider them in the revision of your manuscript.

Reviewers' comments:

Reviewer's Responses to Questions

**Comments to the Author**

1. Is the manuscript technically sound, and do the data support the conclusions?

Reviewer #1: Yes

Reviewer #2: Partly

2. Has the statistical analysis been performed appropriately and rigorously? 

Reviewer #1: N/A

Reviewer #2: I Don't Know

3. Have the authors made all data underlying the findings in their manuscript fully available?

Reviewer #1: Yes

Reviewer #2: Yes

4. Is the manuscript presented in an intelligible fashion and written in standard English?

Reviewer #1: Yes

Reviewer #2: Yes

5. Review Comments to the Author

Reviewer #1: In this manuscript, the authors focus on the eigenfunction of an age-structured population model as fitness. The function generates an equation, called the Hamilton-Jacobi-Bellman equation, that achieves adaptive control of life history in terms of both the presence and absence of the density effect. Further, they introduce a perturbation method that applies the solution of this equation to the long-term logarithmic growth rate of a stochastic structured population model. They adopt this method to realize the adaptive control of heterogeneity for an optimal foraging problem in a variable environment as the analyzable example. The result indicates that the eigenfunction is involved in adaptive strategies under all the listed environments here. The authors reveal that fitness is closely related to the reproductive value. They show that characteristic functions play an important role in population dynamics even in constant and variable environments. Their model shows that heterogeneity is more likely to evolve in a variable environment than in a constant environment. After reading this work, I have found it interesting and technically sound. However, I still have some questions or comments on this work.

(1) In this work, the authors adopt the method to realize the adaptive control of heterogeneity for an optimal foraging problem in a variable environment as the analyzable example. In this analyzable example, I wonder what the meanings of adaptive control are and whether the adaptive control rule makes senses.

(2) There are some typos in the manuscript, e.g., see line 266. I suggest the authors proof-read again and have the spelling and language checked.

Reviewer #2: I must be honest that the topics of the manuscript is rather far from my expertise, and I am not be able to comment on the correctness of the research. Hence I will only comment on some mathematical part and presentation, but can not provide a recommendation and leave it to the editor.

6. PLOS authors have the option to publish the peer review history of their article (what does this mean?). If published, this will include your full peer review and any attached files.

Reviewer #1: No

Reviewer #2: No

---

## [Author Response · Author response to Decision Letter 0]

10 Aug 2021

Reviewer #1: 

In this manuscript, the authors focus on the eigenfunction of an age-structured population model as fitness. The function generates an equation, called the Hamilton-Jacobi-Bellman equation, that achieves adaptive control of life history in terms of both the presence and absence of the density effect. Further, they introduce a perturbation method that applies the solution of this equation to the long-term logarithmic growth rate of a stochastic structured population model. They adopt this method to realize the adaptive control of heterogeneity for an optimal foraging problem in a variable environment as the analyzable example. The result indicates that the eigenfunction is involved in adaptive strategies under all the listed environments here. The authors reveal that fitness is closely related to the reproductive value. They show that characteristic functions play an important role in population dynamics even in constant and variable environments. Their model shows that heterogeneity is more likely to evolve in a variable environment than in a constant environment. After reading this work, I have found it interesting and technically sound. However, I still have some questions or comments on this work.

A. Thank you very much for your interest. We have endeavored to address your questions and comments both here and in the revised manuscript.

(1) In this work, the authors adopt the method to realize the adaptive control of heterogeneity for an optimal foraging problem in a variable environment as the analyzable example. In this analyzable example, I wonder what the meanings of adaptive control are and whether the adaptive control rule makes senses.

A. To clarify the meaning of adaptive control in the analyzable example, we have added a sentence stating that the control provides the optimal size growth curve maximizing the long term logarithmic growth rate (line 426 in the revised manuscript).

To show that the analyzable model is an example of our framework, we have added the description that Eq. (68) is one of the HJB equations (Eq. (35)) (line 430), and that the derivation of Eq. (68) from the reproductive value is achieved in the same way as the general HJB equation (under line 430).

(2) There are some typos in the manuscript, e.g., see line 266. I suggest the authors proof-read again and have the spelling and language checked.

A. Thank you for pointing out those oversights. We have corrected all typos and had the revised manuscript proofread by a professional English language editing service.

Reviewer’s comments in the attachment

1. What is the meaning of Pt(a; y) and the meaning of equation (3)? Please provide interpretation for these. 

A. Thank you very much for your comments. Accordingly, we have added the following interpretations to line 115: 

“Let the population vector $P_{t}\\left(a, y\\right)$, in which each individual follows the ingredients Eq. \\eqref{sde0}, $F\\left(a,y\\right)$, and Eq.\\eqref{dr0}, be a cohort density at age $a$ at a state $y$ in time $t$.” and “Eq. \\eqref{mac1} implies that the cohort transitions dynamically for age $a$ and state $y$ at time $t$.” 

2. line 117, the citation [16] should be placed after the sentence in line 116.

 A. We agree with you; the corresponding change has been made.

3. In (4){line 119: the integral variable y should be replaced by another one since it confuses

with y on the LHS.

 A. We agree with you; the corresponding change has been made.

4. line 123, p(t; a) := P(t; a_): the notation P(t; a; _) is not consistent with the previous notation

Pt(a; _) (the time variable t is in the subscript).

A. Thank you very much for pointing this out. The symbol has been changed in the revised manuscript.

5. line 125, what is the reference "eqrefmac2"? there should be a blank space after the bracket

there.

A. We have corrected the misrepresentation.

6. line 229, "determined in ??.": please replace ?? by a correct reference.

 A. Thank you for pointing this out. We have replaced ?? with the correct reference, S.1. 

7. line 266, "(see ??).": insert a correct reference.

 A. Thank you for pointing this out. We have replaced ?? with the correct reference, S.2.

8. line 157: the full stop punctuation (after (11)) should be removed.

 A. Thank you for pointing this out. We have removed it and moved the equation to line 147.

9. Line 142, please provide a reference for the statement.

 A. Thank you for pointing this out. We have added a relevant reference (Metz and Diekmann 1986).

10. What is r0 in Eq.(12)?

 A. Thank you for this question. We have added its corresponding explanation immediately after Eq. (12).

11. line 229, "determined in ??.": what is ??

 A. Thank you for this question. We have replaced ?? with the correct reference, S.1. 

12. a comma is redundant at the end of Eq (31).

 A. Thank you for pointing this out. We have removed it.

13. the full stop in Eq (32) should be removed.

 A. Thank you for pointing this out. We have replaced it with a comma.

14. After Eq (34), "See ??": what is ??

A. We have replaced ?? with the correct reference, S.2.

15. Please explain how to obtain (41) since it is not trivial at all.

 A. Indeed, there was a lack of explanation for the derivation. Therefore, we explain the operation symbol after new Eq. 41 appears and the requirement formula that is the main result in [22] after Eq. 42 in the revised manuscript. 

 Please check whether they are satisfactory.

---

## [Decision Letter · Decision Letter 1]

1 Sep 2021

Evolution of heterogeneity under constant and variable environmets

PONE-D-21-12079R1

Dear Dr. Oizumi,

We’re pleased to inform you that your manuscript has been judged scientifically suitable for publication and will be formally accepted for publication once it meets all outstanding technical requirements.

Kind regards,

The Anh Han, Ph.D.

Academic Editor

PLOS ONE

Additional Editor Comments (optional):

Reviewers' comments:

Reviewer's Responses to Questions

**Comments to the Author**

1. If the authors have adequately addressed your comments raised in a previous round of review and you feel that this manuscript is now acceptable for publication, you may indicate that here to bypass the “Comments to the Author” section, enter your conflict of interest statement in the “Confidential to Editor” section, and submit your "Accept" recommendation.

Reviewer #1: All comments have been addressed

Reviewer #2: All comments have been addressed

2. Is the manuscript technically sound, and do the data support the conclusions?

Reviewer #1: Yes

Reviewer #2: Yes

3. Has the statistical analysis been performed appropriately and rigorously? 

Reviewer #1: N/A

Reviewer #2: Yes

4. Have the authors made all data underlying the findings in their manuscript fully available?

Reviewer #1: Yes

Reviewer #2: Yes

5. Is the manuscript presented in an intelligible fashion and written in standard English?

Reviewer #1: Yes

Reviewer #2: Yes

6. Review Comments to the Author

Reviewer #1: In the revised manuscript, the authors have addressed my comments accordingly, and I would like to recommend the publication of the work in PLOS ONE.

Reviewer #2: In the revised version, the authors have addresssed all the issues that I raised. Therefore, I recommend to accept the manuscript.

7. PLOS authors have the option to publish the peer review history of their article (what does this mean?). If published, this will include your full peer review and any attached files.

Reviewer #1: No

Reviewer #2: No

---

## [Editor Report · Acceptance letter]

2 Sep 2021

PONE-D-21-12079R1 

Evolution of heterogeneity under constant and variable environments 

Dear Dr. Oizumi:

I'm pleased to inform you that your manuscript has been deemed suitable for publication in PLOS ONE. Congratulations! Your manuscript is now with our production department. 

Kind regards, 

on behalf of

Dr. The Anh Han 

Academic Editor

PLOS ONE